# The *Helicobacter pylori* Urease Virulence Factor Is Required for the Induction of Hypoxia-Induced Factor-1α in Gastric Cells

**DOI:** 10.3390/cancers11060799

**Published:** 2019-06-10

**Authors:** Manuel Valenzuela-Valderrama, Paulina Cerda-Opazo, Steffen Backert, María Fernanda González, Nicolás Carrasco-Véliz, Carla Jorquera-Cordero, Sergio Wehinger, Jimena Canales, Denisse Bravo, Andrew F. G. Quest

**Affiliations:** 1Laboratorio de Microbiología Celular, Instituto de Innovación e Investigación en Salud, Facultad de Ciencias de la Salud, Universidad Central de Chile, Santiago 8320000, Chile; manuel.valenzuela@ucentral.cl (M.V.-V.); nicolas.carrasco.veliz@gmail.com (N.C.-V.); 2Centro de Estudios Avanzados en Enfermedades Crónicas (ACCDiS), Independencia, Santiago 8380000, Chile; denisseb@gmail.com; 3Laboratorio de Comunicaciones Celulares, Centro de estudios en Ejercicio, Metabolismo y Cáncer (CEMC), Instituto de Ciencias Biomédicas (ICBM), Facultad de Medicina, Universidad de Chile, Independencia, Santiago 8380000, Chile; paulina.cerda.opazo@gmail.com (P.C.-O.); mfe.gonzalez@gmail.com (M.F.G.); carlajorquera@gmail.com (C.J.-C.); jim.canales.u@gmail.com (J.C.); 4Department of Biology, Division of Microbiology, University of Erlangen-Nuremberg, 91054 Erlangen, Germany; steffen.backert@fau.de; 5Thrombosis Research Center, Medical Technology School, Department of Clinical Biochemistry and Immunohaematology, Faculty of Health Sciences, Interdisciplinary Excellence Research Program on Healthy Aging (PIEI-ES), Universidad de Talca, Talca 3460000, Chile; stevefoxster@gmail.com; 6Laboratorio de Microbiología Oral, Departamento de Patología y Medicina Oral, Facultad de Odontología, Universidad de Chile, Independencia, Santiago 8380000, Chile

**Keywords:** *Helicobacter pylori*, urease, HIF-1α, Toll-like receptor 2 (TLR2)

## Abstract

Chronic *Helicobacter pylori* infection increases the risk of gastric cancer and induction of hypoxia-induced factor (HIF), which is frequently associated with the development and progression of several types of cancer. We recently showed that *H. pylori* activation of the PI3K-AKT-mTOR pathway in gastric cells increased HIF-1α expression. Here, we identified the *H. pylori* virulence factor responsible for HIF-1α induction. A mutant of the *H. pylori* 84-183 strain was identified with reduced ability to induce HIF-1α. Coomassie blue staining of extracts from these bacteria separated by sodium dodecyl sulfate polyacrylamide gel electrophoresis (SDS-PAGE) revealed poor expression of urease subunits that correlated with reduced urease activity. This finding was confirmed in the 26695 strain, where urease mutants were unable to induce HIF-1α expression. Of note, HIF-1α induction was also observed in the presence of the urease inhibitor acetohydroxamic acid at concentrations (of 20 mM) that abrogated urease activity in bacterial culture supernatants, suggesting that enzymatic activity of the urease is not required for HIF-1α induction. Finally, the pre-incubation of the human gastric adenocarcinoma cell line AGS with blocking antibodies against Toll-like receptor-2 (TLR2), but not TLR4, prevented HIF-1α induction. In summary, these results reveal a hitherto unexpected role for the urease protein in HIF-1α induction via TLR2 activation following *H. pylori* infection of gastric cells.

## 1. Introduction

*Helicobacter pylori* (*H. pylori*) colonizes the human gastric mucosa of approximately half of the human population worldwide [1]. These bacteria are often acquired during childhood and can persist asymptomatically over a lifetime in most cases; however, in some individuals, chronic infection can promote a more exacerbated inflammatory response, which triggers a loss of gastric epithelial cells by apoptosis and an accumulation of epigenetic changes that lead to alterations in signaling pathways, affecting epithelial cell turnover, cell differentiation, and finally gastric epithelial homeostasis [2,3]. *H. pylori* infection is widely considered to increase the risk of developing gastric cancer and is held responsible for approximately 90% of cases of intestinal-type gastric carcinoma [4]. This disease is characterized by a cascade well-defined histological changes in the gastric mucosa, beginning with chronic gastritis, followed by atrophy and intestinal metaplasia, which may ultimately lead to dysplasia and gastric cancer [5,6].

*H. pylori* has developed several strategies in order to colonize and persist in gastric niche conditions [7] where the low pH of the gastric lumen usually acts as a major limitation to bacterial growth. To overcome this, *H. pylori* expresses high levels of urease, an enzyme that catalyzes the hydrolysis of urea to generate ammonia, which buffers the cytoplasm, periplasm, and immediate environment of the bacteria [8]. Furthermore, the bacterial adhesins BabA, SabA or HopQ permit binding of the bacteria to the gastric epithelium, thereby preventing elimination with gastric emptying. Also, *H. pylori* virulence factors are associated with the acquisition of immune tolerance characterized by a protective CD24^+^ CD25^+^ regulatory T-cell response, which precludes clearance of infection [9]. For instance, the vacuolating protein VacA is associated with inhibition of T-cell responses and antigen presentation [10,11]. On the other hand, highly virulent strains express the oncogenic protein CagA, which is associated with exacerbated inflammatory responses, the ability to immortalize primary cells and to promote tumor formation in several animal models [12]. Together, these factors promote carcinogenesis, given that successful infection of the gastric mucosa by *H. pylori* induces inflammation, cell death, and the loss of cellular homeostasis due to the epigenetic alterations triggered by chronic exposure to reactive oxygen and nitrogen species [13]. In this context, the dysregulation of signaling pathways that favor proliferation and survival of gastric epithelial cells in this noxious environment is crucial to the development of cancer.

The importance of *H. pylori* urease for infection has been corroborated in animal models showing that mutant strains lacking the enzyme are unable to persistently colonize the gastric mucosa [14]. It is important to note that on the one hand, the enzyme is surface-exposed [15], and on the other hand, it is present in large amounts within the cytoplasm. Thus, to reach the latter pool of the enzyme, the urea must cross the outer and inner membranes and enter the bacterial cytoplasm with the help of a specific channel (UreI) localized in the inner membrane [16]. The urease is formed by the major subunits UreA and UreB of 27 and 64 kDa, respectively. These subunits are encoded by the urease operon [17] but require other accessory proteins for the correct assembly of the subunits and incorporation of two essential Ni^2+^ ions into the active site [18]. Here, it is important to mention that mutant strains lacking the UreI channel cannot resist the stress of low pH in the stomach even if urea is present in the extracellular milieu [19]. Of note, a fraction of the urease enzyme is known to be associated with the external leaflet of the inner membrane, and liberation may occur via secretion [20,21] or a mechanism of autolysis [22,23]. The existence of such “extracellular” urease relates to functions that go beyond those of an enzyme and have been connected to processes of inflammation [24], immune tolerance [25], angiogenesis [26] and persistence in the gastric niche [15].

The human transcription factor hypoxia-induced factor-1 (HIF-1), is a heterodimer formed by the subunits α and β [27]. While the β subunit is constitutively expressed, the α subunit only increases under conditions of hypoxia, because, under normoxic conditions, it is sequestered by a cytoplasmic complex that targets the protein for degradation via the proteasome pathway [27]. In hypoxia, HIF-1α is liberated from the complex and no longer degraded, such that cytoplasmic protein levels of this subunit increase, thereby permitting the formation of a heterodimer with the β-subunit that translocates to the nucleus to promote transcription of a large number of genes implicated in the maintenance of homeostasis. This response allows the cells to adapt metabolically to the hypoxic conditions and also to promote angiogenesis to improve tissue irrigation by blood vessels [27]. However, non-canonical roles for HIF-1α have also been described, for instance, relating HIF-1α induction to cell cycle arrest [28]. Additionally, HIF-1α is frequently observed during the development and progression of many types of cancer, including gastric cancer [29]. Interestingly, a recent study by our group showed that *H. pylori* infection of gastric cells induced HIF-1α expression in normal oxygen conditions by transient activation of the PI3K/AKT/mTOR pathway [30]. Surprisingly, however, this increase was not linked to increased expression of target genes such as LDH, VEGF, and GLUT1, but rather to changes in the cell cycle related to decreases in Cyclin D1 [30], in agreement with non-canonical roles ascribed to HIF-1α by others [28]. Nevertheless, in our previous study, we did not identify the bacterial virulence factor implicated in triggering this response.

Currently, it is not known whether the *H. pylori* urease participates in the induction of HIF-1α in normoxic conditions; however, the high prevalence of this enzyme, together with its documented ability to activate some signaling pathways, made it an interesting candidate. In this study, we present evidence showing that the *H. pylori* urease is responsible for HIF-1α induction in gastric cells. Importantly, this ability did not involve the enzymatic activity of the urease but rather depended on the induction of Toll-like receptor 2 (TLR2)-dependent signaling. These results provide novel insight to a non-canonical biological function of the *H. pylori* urease that is likely to be highly relevant to understanding how the bacteria triggers inflammation in the host.

## 2. Results

### 2.1. Characterization of the Ability of Helicobacter pylori Mutant Strains to Induce HIF-1α in Gastric Cells

As outlined above, the *H. pylori* virulence factor responsible for HIF-1α induction was not defined in our previously published study. Thus, we tested several strains defective in the production of relevant virulence factors such as GGT, VacA, LPS, or CagA. Among these, a Δ*cagA* mutant derived from the *H. pylori* strain 84-183 showed a remarkably reduced ability to induce HIF-1α in infected gastric cells. In agreement with our previous data [30], the parental strain was able to induce the HIF-1α reporter activity both in human gastric adenocarcinoma AGS (Figure 1A) and MKN45 (Figure 1B) cell lines. However, this response was not observed when cells were infected with the isogenic Δ*cagA* strain (Figure 1A,B). Moreover, we confirmed the absence of CagA expression in the mutant strain via Western blot analysis of total protein lysates (Figure 1C).

Then, to rule out the possibility that a clonal phenomenon, i.e., another mutation, was responsible for the loss of HIF-1α activation in this mutant, a *cagA* isogenic mutant was generated in the *H. pylori* 26695 reference strain by disrupting the wild-type allele with the insertion of a CAT cassette (see material and methods). As shown in Figure 1D, the infection of AGS cells with three different chloramphenicol resistant clones revealed that CagA was not essential for HIF-1α induction as revealed by the Western blot (Figure 1D) and luciferase reporter assays (Figure 1E). Similar responses were observed when parental and isogenic mutants were compared. Thus, we assumed that the *H. pylori* 84-183 Δ*cagA* mutant strain must harbor an additional defect beyond the *cagA* deletion. With this in mind, this strain was further studied in order to identify the element responsible for HIF-1α induction.

### 2.2. The Isogenic cagA mutant H. pylori 84-183 Strain Displayed Reduced Urease Protein Levels and Activity

To gain insight into the possible differences between wild-type *H. pylori* 84-183 and the Δ*cagA* mutant strain, the protein profiles of both were compared after separation by sodium dodecyl sulfate polyacrylamide gel electrophoresis (SDS-PAGE) in 10% gels and staining with Coomassie blue. Notable differences in the abundance of at least three proteins were observed in the total protein lysate (Figure 2A).

Particularly, the protein bands of approximately 130, 65 and 30 kDa appeared to be less abundant in the mutant compared with the parental strain. These bands were excised and analyzed by mass spectrometry (see Appendix A). The results obtained identified that the major bands that were missing were the CagA (130 kDa theoretical MW), UreB (62 kDa theoretical MW), and UreA (26 kDa theoretical MW) proteins. This finding was corroborated by immunodetection of UreB expression by Western blotting and quantification of urease activity. As shown (Figure 2B), UreB protein levels were significantly reduced in the mutant strain when compared with the parental strain (~75% reduction). On the other hand, VacA protein levels, as a control, remained unchanged. Additionally, urease activity was quantitatively determined by measuring ammonia production when total bacterial lysates were incubated with a urea-containing buffer (see material and methods). As shown in Figure 2C, urease activity was significantly reduced in the mutant versus the parental strain. Together, these results confirmed that the Δ*cagA* mutant strain harbored an additional defect that leads to reduced urease protein and activity levels, which could potentially explain the inability of this particular strain to induce HIF-1α.

### 2.3. An Isogenic Urease Mutant of the H. pylori 26695 Strain Reduced the Ability to Induce HIF-1α Protein Expression in AGS Cells

Given that the *H. pylori* urease appears to be involved in several cellular responses beyond its known role in acid resistance and gastric colonization [26,31], and that urease protein levels/activity were reduced in the 84-183 Δ*cagA* mutant, we hypothesized that the urease might be responsible for HIF-1α induction in gastric cells. As illustrated (Figure 3A), the genes encoding the urease subunits A and B are adjacent to one other in the urease operon of the annotated *H. pylori* 26695 genome.

As is indicated, the deletion of a 988 base pair region and insertion of the CAT cassette was accomplished by allelic exchange following chloramphenicol selection (Figure 3A, see material and methods). Analysis of total protein extracts of two resistant clones by SDS-PAGE and staining with Coomassie blue confirmed the absence of the UreA and UreB proteins (Figure 3B, arrows on the right). Additionally, the absence of urease activity was confirmed by a qualitative test consisting of the incubation of live bacteria in a buffer containing urea and the pH indicator phenol red. As expected, protein extracts from the *H. pylori* 26695 Δ*ureA/B* clones 1 and 2 were not able to change the color of the phenol red indicator from yellow to pink, unlike extracts from the parental strain (Figure 3C). As a control, the irreversible urease inhibitor, acetohydroxamic acid (AHA, 20 mM), was included in the reaction. Also, urease activity was determined quantitatively from the protein extracts of mutant and parental strains. As is shown in Figure 3D, the activity in lysates of Δ*ureA/B* mutants was undetectable. Following this characterization, the ability to induce HIF-1α protein levels in AGS cells was tested after 8 h of infection. As shown (Figure 3E), Δ*ureA/B* mutants were not able to induce HIF-1α protein levels, unlike the parental *H. pylori* 84-183 or 26695 strains. Since this difference could have been due to a delay in the kinetics of the response, we carried out an experiment comparing the induction of HIF-1α protein levels in response to parental and *ΔureA/B* clon 1 mutant (Δ*ure1*) strains over a 24 h period. As shown (Figure 3F), the Δ*ureA/B* clon 1 mutant (Δ*ure1*) was not able to induce HIF-1α at any of the time points tested. Thus, the urease virulence factor is required for HIF-1α induction.

### 2.4. Enzymatic Urease Activity Appears Not to Be Required for HIF-1α Induction

Taking into account the multiple roles ascribed to the urease in gastric pathogenesis triggered by *H. pylori* infection, we wondered if enzymatic degradation of urea, and thus enzyme activity, was necessary for HIF-1α induction. With this in mind, we performed infection experiments in the presence of the potent urease inhibitor AHA. Indeed, AHA at concentrations of 20 mM during infection of AGS cells potently blocked enzyme activity (see Figure 3), but did not suppress the ability of *H. pylori* to induce HIF-1α (Figure 4A).

Alternatively, we wondered if urea availability might be a limiting factor for HIF-1α induction. As shown (Figure 4B), the addition of 2 mM urea did not alter substantially the levels of HIF-1α induction in response to *H. pylori*. Deferoxamine (DFO), an iron chelator and potent HIF-1α inducer, robustly increased HIF-1α protein levels. Together, these results suggest that urease activity is not required for the induction of HIF-1α.

### 2.5. TLR2 Is Implicated in the Induction of HIF-1α by the H. pylori Urease

Our results indicated that enzymatic urease activity was not essential for HIF-1α induction. Thus, we explored the possibility that the interaction between the bacterial urease and a host surface molecule, such as a receptor, was required. In this respect, we noted that urease has been shown previously to bind to the Toll-like receptor 2 (TLR2) to trigger inflammasome activation [31,32]. On the other hand, HIF-1α activation has also been related to the production of inflammatory mediators [33]. In order to determine whether TLR2 was relevant to HIF-1α induction by *H. pylori* urease, AGS cells (TLR2+ and TLR4+) [34,35], were incubated with an antibody specific for TLR2 prior to *H. pylori* infection. As shown (Figure 5A), the presence of this TLR2 neutralizing antibody abrogated urease-dependent HIF-1α induction when AGS cells were infected with either the 26695 or the HPAG1 *H. pylori* strains. As a control, incubation with an equivalent amount of neutralizing antibody directed against TLR4 was unable to block HIF-1α induction following *H. pylori* infection. Importantly, the HPAG1 strain has previously been shown to activate TLR2 signaling by a direct association between the bacterial urease and the receptor [36]. Taken together, these observations implicate the involvement of TLR2 in *H. pylori* urease-mediated induction of HIF-1α in infected gastric cells.

## 3. Discussion

In a previous study, we showed that infection of gastric cells with *H. pylori* induces HIF-1α protein levels, which in turn coincided with a decrease in Cyclin D1 half-life and alterations in the cell cycle. However, the bacterial virulence factor involved was not identified. In the present study, we provide evidence that the *H. pylori* urease protein is required, but that activity of the enzyme is not necessary. In doing so, we describe a novel, non-catalytic function for the *H. pylori* urease in the infection of gastric cells.

While testing isogenic *H. pylori* mutants lacking individual virulence factors, we identified a Δ*cagA* mutant strain that was unable to induce HIF-1α. However, upon generating a *cagA* deletion mutant in the reference strain 26695, we observed no change in the ability to induce HIF-1α, suggesting that this factor was not likely to be involved and that the observed response was probably due to a clonal effect. The characterization of the 84-183 Δ*cagA* mutant strain revealed decreased expression of urease subunits that coincided with decreased enzyme activity. Given that previous studies had reported on the isolation of spontaneous mutants lacking urease from a culture of the *H. pylori* strain 60190 [37], and that the 84-183 strain lacking *cagA* had been obtained by the same group, we considered the possibility that the absence of urease might account for the inability of this strain to induce HIF-1α.

The induction of HIF-1α by other human pathogenic bacteria, viruses, fungi, and protozoa has been previously reported, but evidence indicated that such induction would rely on a mechanism dependent on oxygen consumption by the bacteria during co-culture [38,39]. Additionally, the HIF-1α promoter contains response elements for the transcription factor NF-κB [40] that make HIF-1α a target that is inducible by pro-inflammatory events. The important point here is that the induction of HIF-1α by *H. pylori*, unlike what is common for other microorganisms, appears to be due to a mechanism dependent on one particular virulence factor—the urease. However, we cannot entirely rule out contribution from other potentially pro-inflammatory factors such as CagA, LPS metabolites, VacA, etc., that may contribute to a minor extent by activation of NF-κB.

To date, three trans-membrane proteins have been identified that are able to bind to *H. pylori* urease. Urease A has been shown to bind to Major Histocompatibility Complex (MHC) class II molecules, and urease B binds to class II MHC-associated molecules with the invariant chain (Ii) or CD74 [41]. In addition, the interaction between urease B and TLR2 was recently described [31]. In the first two cases, the association of urease with these molecules was linked to the production of pro-inflammatory factors and apoptotic cell death. However, the TLR2–urease interaction has been associated with the generation of immune tolerance responses through the activation of Treg cells in a murine asthma model [31]. In agreement, the TLR2 is considered important for the induction of tolerogenic immune responses following *H. pylori* infection [42]. Consistent with this view, HIF-1 is known to play a key role in the induction of the innate and adaptive immune responses [43]. Also, a connection between TLR2 and HIF-1 has been described during the maturation of dendritic cells [44]. This urease/TLR2/HIF-1 axis has been studied mainly in the context of cells of the immune system and been associated with the generation of tolerance. An interesting emerging question, therefore, is whether activation of this pathway in gastric cells may also participate in the generation of tolerance, or rather it be relevant to the sequence of events leading to pre-neoplastic lesions. Future experiments will address these relevant issues.

The schematic shown in Figure 5B summarizes our current thinking, although some questions related to the induction of HIF-1α through signaling by TLR2 remain to be addressed in future studies. First, it still remains unclear which of the urease subunits is responsible for the activation of TLR2; second, it remains to be determined whether the increase in HIF-1α is due to the stabilization of the protein or due to an increase in the levels of transcription, possibly via NF-kB, since *H. pylori* reportedly activates NF-κB through TLR2 [45]. As described by our laboratory, *H. pylori* induces HIF-1α activity as a transcription factor via a PI3K-dependent pathway to produce G0/G1 cell cycle arrest in epithelial gastric cells [30]. However, other virulence factors, such as LPS, likely contribute to the interaction between the bacterium and the host. Indeed, some authors have reported that *H. pylori* LPS induces TLR2 recruitment to lipid rafts, followed by receptor clustering, to produce an inflammatory response [46]. Also, activation of the PI3K/Akt/mTOR signaling pathway by TLR2 occurs when PI3K is recruited to the cell membrane, indicating the existence of cross-talk between the receptor and PI3K in inflammatory signaling [47]. Thus, we suggest that *H. pylori* urease could induce TLR2 signaling via the PI3K pathway either to induce HIF-1α transcription and/or prevent HIF-1α degradation by the proteasome (Figure 5B).

## 4. Material and Methods

### 4.1. Cell Culture

The gastric cancer cell lines AGS (ATCC CRL-1793) and MKN74 (JCRB0255) were cultured in RPMI-1640 medium (Gibco, Paisley, UK) supplemented with 10% fetal bovine serum (FBS, PAN Biotech, Aidenbach, Bavaria, Germany) and antibiotics (10,000 U/mL penicillin and 10 μg/mL streptomycin) in a controlled atmosphere (5% CO_2_, 70% humidity at 37 °C).

### 4.2. Bacterial Culture

*H. pylori* strain 26695 (700392) was obtained from the ATCC (American Type Culture Collection), 84-183 and 84-183 ΔcagA were provided by Dr. Guillermo Perez-Perez. HPAG1 wildtype and urease mutant strains were provided by Dr. Douglas Berg. All strains were grown on trypticase soy agar (TSA) plates, supplemented with 5% horse donor serum (PAN Biotech), the nutrient supplement—Vitox (Oxoid Limited, Wade Road, Basingstoke, Hampshire, UK)—and the selective supplement—Dent (Oxoid Limited)—for 24 h under low oxygen tension conditions (5% CO_2_, 70% humidity at 37 °C).

### 4.3. Construction of H. pylori CagA and Urease Mutants

Briefly, 5′ and 3′ fragments of approximately 300 base pairs (bp) were amplified using hybrid primers containing 5′ extreme sequences complementary to the chloramphenicol acetyltransferase (CAT) cassette [48]. The mutant allele with CAT insertion was obtained by PCR, and the resulting amplicon was used to transform wild-type bacteria that were selected in TSA plates with chloramphenicol as previously described [49]. Deletion of the respective genes in the resistant colonies was confirmed by Western blotting using specific antibodies. VacA was employed as a positive control (for antibody origin and dilutions, see 4.7 Western blotting).

### 4.4. Experiments with TLR2 and TLR4 Blocking Antibodies

The human gastric adenocarcinoma cell line AGS (ATCC CRL-1739™) was cultured in RPMI1640 medium supplemented with 10% fetal calf serum (Gibco, Paisley, UK) and a penicillin/streptomycin cocktail (Sigma-Aldrich, Merck KGaA, Darmstadt, Germany). The cells were grown at 37 °C and 5% (*v/v*) CO_2_ in 6-well plates. When the cells reached ~70% confluency, they were washed two times with phosphate buffer saline (PBS), and fresh RPMI-1640 medium without antibiotics was added. Thirty minutes prior to infection, 4 µg/mL TLR2 neutralizing antibody (InvivoGen, San Diego, CA, USA), or 4 μg/mL TLR4 neutralizing antibody (InvivoGen) as a control, was added as described [50]. Infections with the above *H. pylori* strains were performed at a Multiplicity of Infection (MOI) of 50 for 12 h.

### 4.5. Infection of Gastric Cells

Gastric cells (1 × 10^6^) were seeded onto 60 mm diameter plates (BD Falcon, BD Biosciences, Billerica, MA, USA) and cultured for 24 h. Before infection, the medium was substituted by RPMI-1640 (Gibco) supplemented with FBS without antibiotics. For infection, bacteria were collected in PBS, centrifuged at 4000× *g* and resuspended again in PBS. An absorbance of 0.4 units at 560 nm was considered equivalent to 3 × 10^8^ bacteria. Cells were infected with a multiplicity of infection (MOI) of 1:100, as previously described [13], except for the experiments shown in Figure 5, where an MOI of 50 was used.

### 4.6. Luciferase Reporter Assay

Gastric cells were transfected with a luciferase reporter plasmid containing the HRE response elements (pGL3-HRE plasmid) and a control plasmid constitutively expressing β-galactosidase (pON plasmid) using the transfection agent Viafect (Promega, Madison, WI, USA), following the instructions provided by the manufacturer, as described previously [30]. After 16 h of transfection, cells were infected with *H. pylori* (MOI 100) for 8 h and then lysed in a buffer containing 100 mM KH_2_PO_4_ (pH 7.9), 0.5% Triton X-100, and 1 mM DTT. Protein extracts were centrifuged at 11,000× *g* for 3 min and supernatants were used to determine luciferase and β-galactosidase activities in a multiplate reader Infinite M200Pro (TECAN, Männedorf, Switzerland).

### 4.7. Western Blotting

Total protein extracts were obtained from cellular or bacterial pellets in a buffer containing 20 mM HEPES (4-(2-hydroxyethyl)-1-piperazineethanesulfonic acid, pH 7.4), a cocktail of protease inhibitors (12.5 μg/mL leupeptin, 10 μg/mL antipain, 100 μg/mL benzamidine, 1 mM phenylmethylsulphonyl fluoride, 1 mM sodium orthovanadate, and 10 mM sodium fluoride), 0.1% SDS and 0.05% NP-40 (IGEPAL) by sonication. Samples were centrifuged at 11,000× *g* and supernatants were collected and stored at –20 °C. Proteins (80 μg) were separated by SDS-PAGE (10%) and transferred to nitrocellulose membranes, as described previously [13]. Membranes were blocked with PBS containing 0.1% Tween and 5% skimmed milk for 1 h and then with primary antibodies diluted in the same buffer: anti-human HIF-1α (1:1000, BD Bioscience, MA, USA, # 610958), anti-CagA (1:3000, Santa Cruz Biotechnology, Dallas, TX, USA), #sc-28368), anti-VacA (1: 1000, Santa Cruz Biotechnology #sc-32746), anti UreB (1:25,000, Abcam, Cambridge, MA, USA), #ab127916), and an anti-human β-actin (1:15,000, Sigma #A5316). Primary antibodies were detected with secondary antibodies conjugated with Horseradish Peroxidase (HRP) and the EZ-ECL kit (Biological Industries, Cromwell, CT, USA). Images were captured by a Hamamatsu Digital Camera C8484. Protein levels were quantified by densitometric analysis using the software UnScan-it v. 6.1 (Orem, UT, USA).

### 4.8. Determination of Urease Activity

Urease activity was determined as described by Weatherburn [51]. Briefly, a suspension of bacteria (0.4 U absorbance at 560 nm) was lysed by sonication in lysis buffer (50 mM HEPES and 1 mM EDTA, pH 7.5). Then, the protein concentration was adjusted to 1 µg of protein/50 µL lysis buffer. This volume was added to 1950 µL lysis buffer containing 25 mM urea. Subsequently, 50 µL aliquots were removed every 30 min, immediately mixed with 700 µL of a solution containing phenol and sodium nitroprusside, and then with 700 µL of solution B (NaOH and sodium hypochlorite). The mixture was incubated for 30 min at 37 °C until a blue color appeared. Absorbance was determined at 625 nm. Ammonia production was determined by comparing values obtained in samples with a standard curve generated using NH_4_Cl. Values were expressed as mmol NH_4_/min/µg protein. Alternatively, urease activity was visualized by incubating bacterial suspensions with Brucella broth containing urea (5 mM) and red phenol as a pH indicator. As a control, the urease inhibitor acetohydroxamic acid (AHA, Sigma) was added at a final concentration of 20 mM.

### 4.9. Statistical Analysis

Data were processed using the software Instat, version 3.05 (GraphPad Software, San Diego, CA, USA; available in: http://www.graphpad.com). Numerical data are shown as the means ± standard error of the mean (SEM). Statistical significance of differences was determined using the Wilcoxon-Mann-Whitney test for non-parametric data and were considered significant at *p* < 0.05.

## 5. Conclusions

Our findings identify a non-enzymatic role for *H. pylori* urease by showing that it acts as a ligand to induce HIF-1α activation via TLR2 signaling in gastric cells.

## Figures and Tables

**Figure 1 cancers-11-00799-f001:**
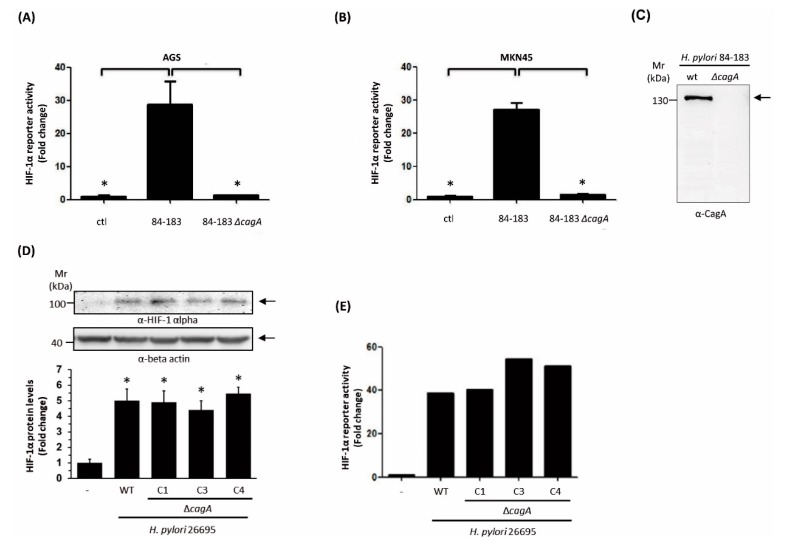
Hypoxia-induced factor (HIF-1) induction in response to *H. pylori* strains in gastric epithelial cells. AGS (**A**) and MKN45 (**B**) cells were transfected with the reporter plasmid pGL3-HRE (HIF reporter) and the control plasmid pON (beta-galactosidase). After 16 h, cells were infected with the *H. pylori* wild-type strain 84-183 or its isogenic Δ*cagA* mutant strain at a Multiplicity of Infection (MOI) 100 for 8 h. Then, protein extracts were prepared to determine luciferase reporter activity. Bars represent values of luminescence normalized to β-galactosidase activity compared with the control condition (mean ± SEM, *n* = 3, * *p* ≤ 0.05). (**C**) Immunodetection of CagA in protein extracts from *H. pylori* parental 84-183 and Δ*cagA* mutant strains. AGS cells were infected with *H. pylori* 26695 and three different Δ*cagA* isogenic clones (C1, C3, and C4) at MOI 100 for 8 h. (**D**) HIF-1α protein and β-actin levels were analyzed by Western blotting. A representative blot is shown (mean ± SEM, *n* = 3, * *p* ≤ 0.05). (**E**) HIF-1 activation was determined by the firefly luciferase reporter assay. A representative result with the three clones is shown.

**Figure 2 cancers-11-00799-f002:**
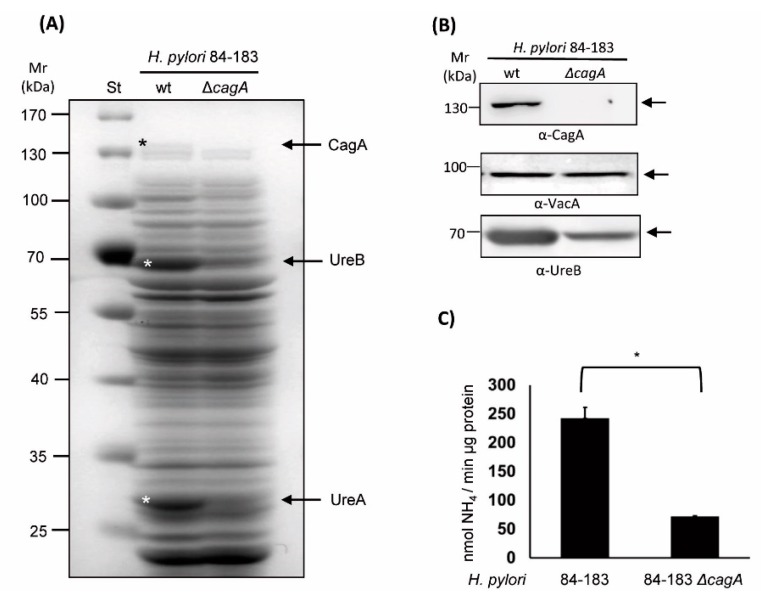
*H. pylori* 84-183 Δ*cagA* mutant revealed reduced urease expression levels. (**A**) Total protein extracts (80 µg) from *H. pylori* 84-183 parental and isogenic mutant strains were separated by SDS-PAGE (10%) and stained with Coomassie blue. A representative gel is shown. Asterisks and arrows indicate the positions of the urease subunits A and B, as well as the cytotoxin CagA (confirmed by mass spectrometry). To the left, molecular weight standards are indicated (130–25 kDa). A representative Western blot showing the absence of CagA, as well as reduced presence of the Urease A and B subunits in the *H. pylori* 84-183 mutant strain. VacA presence was analyzed as a loading control. (**B**) Immunodetection of UreB expression and (**C**) determination of urease activity in protein extracts from 84-183 wild type (wt) and Δ*cagA* mutant strains. Bars are indicative of urease activity determined as NH_4_/min/µg protein (means ± SEM, *n* = 3, * *p* ≤ 0.05).

**Figure 3 cancers-11-00799-f003:**
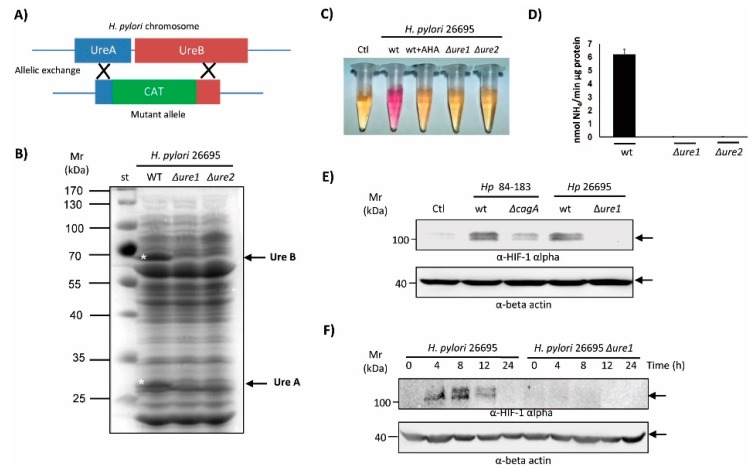
Characterization of the urease mutant in the *H. pylori* 26695 strain. (**A**) Schematic indicating how the urease mutant (Δ*ureA/B*) in the *H. pylori* 26695 strain was obtained by allelic exchange. (**B**) Protein extracts (80 µg) from *H. pylori* 26695 and resistant Δ*ureA/B* clones 1 and 2 were separated by SDS-PAGE (10%) and then stained with Coomassie blue. Asterisks and arrows indicate the positions of UreA and UreB proteins, and the molecular weight markers (10–100 kDa) are shown to the left. A representative gel is shown. (**C**) A suspension of either parental bacteria or Δ*ureA/B* clones 1 and 2 (Δ*ure1* and Δ*ure2*) were added to Brucella broth supplemented with urea 5 mM and the pH indicator red phenol. As a control of the reaction, the urease inhibitor acetohydroxamic acid (AHA, 20 mM) was included. A representative assay is shown. (**D**) Quantitative determination of urease activity in protein extracts from the *H. pylori* 26695 parental strain and its Δ*ureA/B* mutants, clones 1 and 2 (Δ*ure1,* Δ*ure2*). Values are expressed as moles of NH_4_/min/µg protein (means ± SEM, *n* = 3, *p* ≤ 0.05). (**E**) AGS cells were infected with *H. pylori* parental 84-183 and 26695 strains, or their Δ*cagA* and Δ*ureA/B* clon 1 (Δ*ure1*) mutants, respectively, at MOI 100 for 8 h. Protein extracts were prepared and separated by SDS-PAGE on 10% gels. HIF-1α and β-actin protein levels were detected by Western blotting. A representative blot is shown. (**F**) AGS cells were infected for 24 h with *H. pylori* 26695 or the Δ*ureA/B* mutant clone 1 (Δ*ure1*) at MOI 100. Protein extracts were prepared and separated by SDS-PAGE. HIF-1α and β-actin protein levels were detected by Western blotting. A representative blot is shown.

**Figure 4 cancers-11-00799-f004:**
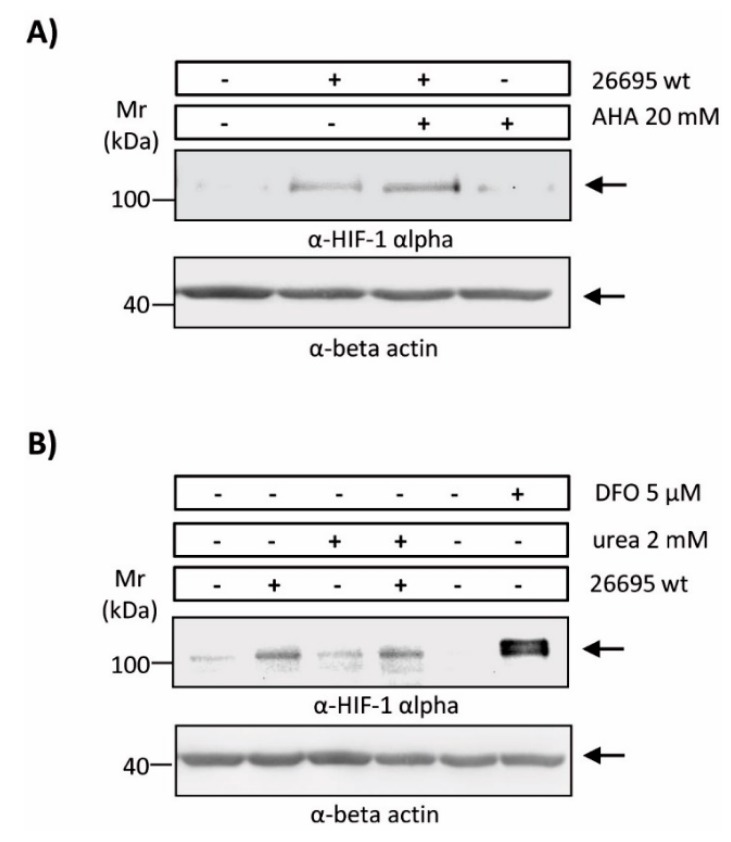
Induction of HIF-1α by *H. pylori* in the presence of AHA or urea in gastric epithelial cells. AGS cells were infected with *H. pylori* 26695 at MOI 100 for 8 h. Protein extracts were prepared and HIF-1α and β-actin protein levels were analyzed by Western blotting. Experiments were performed in the presence/absence of (**A**) the urease inhibitor AHA 20 mM or (**B**) urea 2 mM. Deferoxamine (5 µM) was included as a positive control for HIF-1 induction in normoxic conditions.

**Figure 5 cancers-11-00799-f005:**
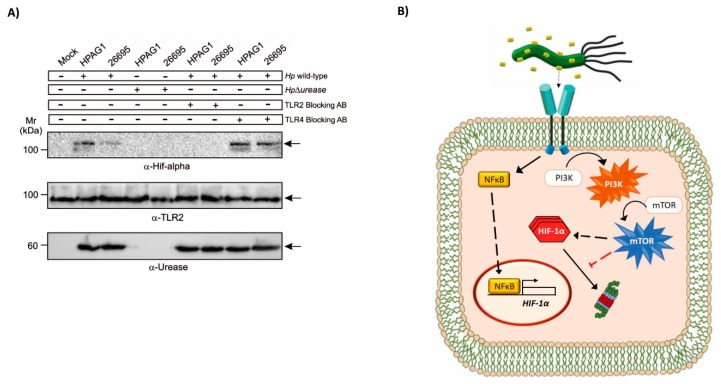
Induction of HIF-1α by *H. pylori* infection is blocked by a TLR2 neutralizing antibody. (**A**) AGS cells were pre-treated with either neutralizing anti-TLR2 or anti-TLR4 antibodies (AB) and then infected with the wild-type *H. pylori* strains 26695 or HPAG1 at MOI 50 for 12 h. Protein extracts of cells were prepared and HIF-1α and β-actin protein levels were analyzed by Western blotting. A representative blot is shown. (**B**) A schematic representation of findings and possible events downstream of the urease-TLR2 interaction is presented. Urease is shown as yellow dots surrounding *H. pylori* and the TLR2 is visible as a blue dimer on the surface of a gastric cell.

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
