# Peer review of "The Helicobacter pylori Urease Virulence Factor Is Required for the Induction of Hypoxia-Induced Factor-1α in Gastric Cells"

_cancers, 2019, doi:10.3390/cancers11060799_

Round 1

Reviewer 1 Report

The manuscript by Valenzuela-Valderrama and colleagues entitled "The Helicobacter pylori urease virulence factor is 3 required for the induction of Hypoxia-induced factor-1α in gastric cells" provides novel insights on the role of the urease enzyme as the virulence factor able to induce the expression of HIF-1α upon infection of gastric cells. Starting from a previously published work by the same group, these investigators show that in the absence of UreA/B urease subunits there is not accumulation of HIF-1α upon infection of AGS cells, and that this effect seems to be related to urease recognition by TLR2 rather than by urease enzymatic activity.

This manuscript is technically sound, represents a large amount of work and provides novel insights that are really interesting for the scientific community. The data presented are well performed, however there are several controls that should be performed or presented in order to improve the manuscript. Moreover, the quality of some experiments should be improved (see below).

Major issues:

Figure 2A shows an SDS-PAGE of total protein extracts from H. pylori 84-183 wt and cagA mutant strains in which the reader should appreciate the decrease of some bands indicated by the authors on the right of the figure. While for UreB this decrease of intensity is clear, for UreA is less evident and for CagA the band is too faint to be appreciated. The quality of this gel must be improved, maybe running a gradient gel and/or staining protein bands with a more sensitive method. 

In the Figure legend of Fig. 2A, the authors say that they load 80 micrograms of total protein extract (line 153, page 4): are you sure? It seems to me a huge amount of protein per lane, please check.

Figure 3B: again, it is hard to appreciate the absence of UreA band in the generated mutant strain from the SDS-PAGE presented. See point 1 above.

The quality and bands' resolution of Western Blots presented in Figure 3E and in Figure 3F are quite low, especially if compared to Western Blots presented in Figures 4 and 5. While in these latter figures HIF-1α band is sharp and clear, bands in Figure 3E and especially 3F are very smeared. These panels should be improved.

The main message of this paper is that urease enzyme (one or both A and B subunits) is the factor that upon contact with gastric cells induces the accumulation of HIF-1α, without the requirement of its enzymatic activity. I bieleve that to fully convince the reader, the ureAB knockout mutant H. pylori 26695 strain must be complemented with a wild type and a catalytically inactive isoform of the enzyme (for example with point mutations in the Ni-binding His residues). The restoration of HIF-1α induction in both complementing strains would strongly sustain the conclusions drawn by the authors about urease (non catalytic) involvement in HIF-1α induction, supporting also the latest part of the work.

Paragraph 2.5 and Figure 5: why was HPAG strain included in this part of the work. Please discuss in the Results section.

Minor issues:

Abstract, line 30 "...in the 26695 strains, where urease mutants was unable to induce..." please check.

Page 6, line 200 H. pylori 26695 and not 26697

Page 6, line 214 Δure in italic

Materials and Methods, paragraph 4.7 Western Blotting: a description/source of the anti-UreB antibody is missing, please add.

Author Response

Reviewer 1:

This manuscript is technically sound, represents a large amount of work and provides novel insights that are really interesting for the scientific community. The data presented are well performed, however, there are several controls that should be performed or presented in order to improve the manuscript. Moreover, the quality of some experiments should be improved (see below).

Response: We thank the reviewer for the positive appreciation of our manuscript.

Major concerns:

1)    Figure 2A shows an SDS-PAGE of total protein extracts from H. pylori 84-183 wt and cagA mutant strains in which the reader should appreciate the decrease of some bands indicated by the authors on the right of the figure. While for UreB this decrease of intensity is clear, for UreA is less evident and for CagA the band is too faint to be appreciated. The quality of this gel must be improved, maybe running a gradient gel and/or staining protein bands with a more sensitive method.

Response: We have improved the resolution as requested by running 10% gels for somewhat longer, thereby permitting better resolution of the bands of interest. These are now clearly visible. In addition, we improved the visibility of CagA, UreA and UreB bands for the reader by adding black or white asterisks in the corresponding lanes.

2)    In the Figure legend of Fig. 2A, the authors say that they load 80 micrograms of total protein extract (line 153, page 4): are you sure? It seems to me a huge amount of protein per lane, please check.

Response: We have checked protocols again and indeed 80 ug of total protein were loaded per lane.

3)    Figure 3B: again, it is hard to appreciate the absence of UreA band in the generated mutant strain from the SDS-PAGE presented. See point 1 above.

Response: see the response to point 1. As pointed out above, we improved the visibility for the reader by adding white asterisks for UreA and UreB bands in the corresponding lanes.

4)    The quality and bands' resolution of Western Blots presented in Figure 3E and in Figure 3F are quite low, especially if compared to Western Blots presented in Figures 4 and 5. While in these latter figures HIF-1α band is sharp and clear, bands in Figure 3E and especially 3F are very smeared. These panels should be improved.

Response: These experiments have been repeated, and we include here images in which the bands of HIF-1α are now clearly visible on the provided blots.

5)    The main message of this paper is that urease enzyme (one or both A and B subunits) is the factor that upon contact with gastric cells induces the accumulation of HIF-1α, without the requirement of its enzymatic activity. I believe that to fully convince the reader, the ureAB knockout mutant H. pylori 26695 strain must be complemented with a wild type and a catalytically inactive isoform of the enzyme (for example with point mutations in the Ni-binding His residues). The restoration of HIF-1α induction in both complementing strains would strongly sustain the conclusions drawn by the authors about urease (noncatalytic) involvement in HIF-1α induction, supporting also the latest part of the work.

Response: We feel the data presented suffice to implicate the Urease virulence factor in the observed effects. First, we observed that Urease A and B subunits are missing in a Hp strain that cannot induce HIF1α. Second, we show that deleting the urease gene in the Hp reference strain 26695 leads to the same effect, that is the Hp cannot induce HIF1a. Third, we show that inhibiting the urease enzyme activity does not ablate the ability to induce HIF1a, suggesting that the protein is acting as a ligand rather than an enzyme in this context. Finally, we provide proof for this novel concept by showing that urease knock-down in another Hp strain using a different deletion strategy eliminates the ability of Hp to induce HIF1a. Furthermore, we show that induction of HIF1a is ablated by neutralizing antibodies against TLR2 but not TLR4. All these results are consistent with the notion that the Hp urease is responsible for the effect and is doing so as a ligand for TLR2 rather than as an enzyme.

Complementation experiments are highly time-consuming and would require in this case first cloning the urease gene and then introducing point mutations that eliminate the activity. The problem with this approach is that a mutation can also lead to non-desirable (non-specific) side effects. Thus, to be sure that the overall structure is maintained, the mutant proteins would have to be characterized structurally. We feel this essentially represents a completely new project and is beyond the scope of what is possible in order to respond within the rather limited period of time available.

6)    Paragraph 2.5 and Figure 5: why was HPAG strain included in this part of the work. Please discuss in the Results section.

Response: This part of the study was done by our collaborator Steffen Backert in Germany. In order to generate the data shown in Fig.5A, a few strains were tested. Best results were obtained for 26695 and HPAG1. HPAG1 is a sequenced Swedish strain (Proc Natl Acad Sci U S A. 2006 Jun 27;103(26):9999-10004). They also observed in HEK293-TLR2 reporter cells that these two strains activate TLR2; this unpublished data is now mentioned in the text.

In any case, we feel that the fact that results were repeated by another group using a different Hp strain adds robustness to the findings. This point is now addressed in the Results section of the resubmitted manuscript.

Minor concerns:

1)    Abstract, line 30 "...in the 26695 strains, where urease mutants was unable to induce..." please check.

Response: this has been corrected as requested

2)    Page 6, line 200 H. pylori 26695 and not 26697

Response: this has been corrected as requested

3)    Page 6, line 214 Δure in italic

Response: this has been corrected as requested

4)    Materials and Methods, paragraph 4.7 Western Blotting: a description/source of the anti-UreB antibody is missing, please add

Response: this has been corrected as requested

English language and style

(x) English language and style are fine/minor spell check required 

Response: the resubmitted manuscript has been carefully revised.

Reviewer 2 Report

The manuscript  “The Helicobacter pylori urease virulence factor is required for the 
induction of Hypoxia-induced factor-1α in gastric cells” by Manuel Valenzuela-Valderrama, et al. is very innovative and well written. The results of the study reveal a new and interesting role for the urease protein in HIF1α induction through TLR2 activation afeter H. pylori infection of gastric cells.These findings
are likely to be highly relevant to understanding how the bacteria triggers inflammation in the host .

Author Response

The manuscript  “The Helicobacter pylori urease virulence factor is required for the 
induction of Hypoxia-induced factor-1α in gastric cells” by Manuel Valenzuela-Valderrama, et al. is very innovative and well written. The results of the study reveal a new and interesting role for the urease protein in HIF1α induction through TLR2 activation after H. pylori infection of gastric cells. These findings are likely to be highly relevant to understanding how the bacteria triggers inflammation in the host.

Response: We thank this reviewer for the very positive and encouraging assessment of our manuscript.

English language and style

(x) Extensive editing of English language and style required 

Response: The resubmitted manuscript has been carefully revised by a native speaker.

Reviewer 3 Report

The authors demonstrated a role for the  urease protein in HIF1α induction via TLR2 activation following H. pylori infection of gastric cells.

The methods are adequate.

The results justify the conclusions drawn.

Author Response

The authors demonstrated a role for the urease protein in HIF1α induction via TLR2 activation following H. pylori infection of gastric cells.

The methods are adequate.

The results justify the conclusions drawn.

Response: We thank this reviewer for the encouraging assessment of our manuscript.

English language and style

(x) Extensive editing of English language and style required 

Response: The resubmitted manuscript has been carefully revised by a native speaker.

Round 2

Reviewer 1 Report

After carefully reviewing the revised version, I believe the manuscript has been significantly improved and now warrants publication in Cancers.